# Clinical Performance Evaluation of a Hyaluronic Acid Dental Gel for the Treatment of Traumatic Ulcers in Patients with Fixed Orthodontic Appliances: A Randomized Controlled Trial

**DOI:** 10.3390/bioengineering9120761

**Published:** 2022-12-03

**Authors:** Marco Tremolati, Marco Farronato, Luca Ferrantino, Francesca Rusconi, Giovanni Lodi, Cinzia Maspero

**Affiliations:** 1Department of Biomedical, Surgical and Dental Sciences, School of Dentistry, University of Milan, 20100 Milan, Italy; 2Fondazione IRCCS Cà Granda, Ospedale Maggiore Policlinico, 20100 Milan, Italy

**Keywords:** hyaluronic acid, polyvinylpyrrolidone, orthodontics, fixed appliance, oral ulceration, ulcer

## Abstract

Background: A newly available gel containing hyaluronic acid (HA) and polyvinylpyrrolidone was tested for efficacy on traumatic oral ulcers (TOU) caused by fixed orthodontic appliances. Methods: A double-blind RCT was conducted to test the new gel versus a placebo. According to the sample size calculation, a total of 60 patients were considered sufficient and randomly allocated to one of the two groups out of a pool of 100 total patients who initially agreed to participate in the study. A VAS scale test and lesion measurements at T0, T1, and T2 were performed on the patients. Results: A total of 70 patients developed TOU, with 8 drop-outs; the intergroup comparison showed a statistically significant greater dimension of the lesion in the control group at T2 when compared to the test group. The pain experienced by the patients belonging to the test group was significantly lower than the pain in the patients in the control group Conclusions: Under the limitations of the study, the new formula might provide faster healing with less pain experienced by the patient when compared to a placebo.

## 1. Introduction

Injury to the oral mucosa (Figure 1) during orthodontic treatment is extremely common [1,2]. Patients with fixed orthodontic braces frequently complain about the presence of sharp edges that irritate the labial or buccal mucosa. Indeed, traumatic oral ulceration (TOU) is one of the most frequent side effects of orthodontic treatments. The TOU incidence reported in the literature is between 60% and 81% of patients wearing braces; TOU onset is mainly in the first few weeks of treatment [1,2,3]. About 47% of adults reported that TOUs are the most troublesome aspect of orthodontic treatment, while 29% of adolescents report that ulcers are the second most troublesome aspect of treatment [1,2]. Pain is part of the inflammatory phase, which usually occurs within 24 h of the onset of the injury. During the first two days after the onset of UTO, patients have difficulty eating and tend to self-medicate the lesion [3]. After removal of the damaging agent, TOUs normally heal within 10–14 days [4]. TOUs can limit chewing and speaking functions and have a negative effect on the patient’s quality of life at the beginning of the therapy, thus compromising good compliance during the orthodontic treatment. Few epidemiological studies have been conducted regarding the clinical course of TOU [2,3,4,5].

The application of orthodontic wax on the area of the braces that is causing the mucosal damage seems to have a positive effect on painful symptoms [5].

Different products to relieve the pain of TOUs, either by preventing their formation or accelerating their healing, have been investigated [6,7]. However, some of those products are not available worldwide but only on local markets, and their effectiveness for the treatment of TOUs has not been clinically tested on a large scale [7].

A brand-new gel (BMG0722) (BMG Pharma S.p.a., Milan, Italy) has been proposed for the treatment of TOUs. BMG0722 is based on hyaluronic acid (HA) and polyvinylpyrrolidone (PVP).

HA is a very important component of connective tissues, belonging to the glycosaminoglycan family. The safety of HA and its effectiveness in promoting the wound healing process have been demonstrated, both in the oral cavity and in other body areas, such as foot ulcers in diabetic patients [8]. According to a systematic review, its usage on oral ulcers is promising [9,10].

The topical use of HA gel could be a valid aid for obtaining a faster and less painful healing course for the orthodontic patient who is suffering from ulcers of the oral mucosa, thus also favoring adhesion to the plan of pre-established care.

PVP is a film-forming agent; it plays a pivotal role in producing a protective barrier that can reduce pain and accelerate wound healing [7].

This study aims to evaluate the effect of the combined use of orthodontic wax and BMG0722 gel on the healing of TOUs during orthodontic treatment (test group), compared to the use of orthodontic wax associated with a placebo gel (i.e., without HA and PVP) (control group).

## 2. Materials and Methods

### 2.1. Trial Design

The present trial was designed as a parallel, double-blind, randomized clinical trial. The allocation ratio was 1:1. The protocol of this study was approved by the Ethics Committee of the University of Milan (approval number 117/20), and its execution was conducted per the principles of good clinical practice (ICH/ISO 14155) and the Helsinki Declaration (2008).

### 2.2. Settings

All clinical procedures were carried out by two experienced operators (M.T. and F.R.) at Clinica Palazzo Riva (Via Sebenico 8, Milan, Italy).

### 2.3. Study Population

Patients attending the clinical center for orthodontic therapy were screened for the following selection criteria:(1)aged between 6 and 18 years;(2)a malocclusion that necessitates the use of a fixed orthodontic appliance (rapid palatal expander, slow mandibular expander, partial fixed, traditional fixed one arch appliance, or both arches);(3)good general health (ASA status I);(4)patients with good oral health (FMPS and FMBS < 20%);(5)absence of caries;(6)periodontal health (absence of periodontal pockets > 4 mm with bleeding on probing, excluding a pseudopocket from the trade-in in progress);(7)patients and parents willing and able to cooperate in all aspects of the protocol;(8)written informed consent to treatment signed by parents (when underage) and participation in the clinical trial.

Subjects were not included if any of the following exclusion criteria were present:
(1)use of drugs that could interact with the wound healing process;(2)smokers;(3)previous history of diseases of the mucous membranes of the oral cavity, in particular, recurrent aphthous stomatitis, ulcers, bullous, or erosive diseases;(4)a history of skin or systemic diseases that may also have evidence in the oral cavity;(5)presence of oral ulcerations of unknown etiology, non-traumatic;(6)pregnancy and breastfeeding;(7)a history of hypersensitivity or allergy to the materials or drugs used in the study;(8)presence of auxiliary extraoral appliances, which could cause additional injuries during the treatment.


If they met these criteria, selected patients were informed about the study’s purpose, and if accepted, they were recruited by signing a written informed consent form.

After the application of the orthodontic device, the selected subject could be again excluded if he/she did not experience any TOU during the first four weeks of treatment.

### 2.4. Intervention

Every patient in the study began fixed orthodontic therapy with the use of an orthodontic device. Patients reporting the onset of a TOU within the first 4 weeks of therapy were examined within 24 h of UTO reporting (T0). The lesion was evaluated at T0, recording the following information:

The location (right/left genetic mucosa, right/left hemi-palate, upper/lower labial mucosa, right/left tongue, upper/lower lip), the size (in mm), and the color.

A photo of the lesion at T0 was taken with a calibrated instrument. After the clinical data of the TOU were recorded, patients were assigned a blind code and randomly allocated into two groups: wax and BMG0722 gel (the test group) or wax and placebo gel (the control group). The patient was asked to report by writing in a diary, any change related to the size of the lesion twice a day (morning and evening) for five days and to specify the intensity of pain on a visual analog scale (VAS scale). A transparent, soft, disposable ruler was given to the patient to measure the lesion. The patients were also asked to take a photograph of the lesion with the calibrated instrument on the third (T1) and fifth day (T2) of the appearance of the TOU (Figure 2 and Figure 3). Patients with persistent pain were advised to take paracetamol as directed to relieve the pain.

The number of daily taken painkillers and a final questionnaire relating to the product used was recorded in the diary. The patients were examined two weeks after the visit following the onset of the TOU, and the patient’s diary was withdrawn.

Stopping guidelines

Participants were instructed to contact the clinic (either by making a phone call or by attending in person) in case they experienced any of the following symptoms: redness (erythema), swelling (edema), and flaking (ulcer) after applying the products under study. They were also informed to stop using the product if they experienced any of the above symptoms.

### 2.5. Sample Size Calculation

The sample size was calculated using the main outcome of the study: the size of the lesion on the fifth day after the appearance of a UTO (T2). A difference in lesion size of 0.5 mm, considered clinically relevant, requires a total of 42 patients (21 per group). Considering that, based on data in the literature, at least 60% of patients with a fixed orthodontic appliance develop a UTO; the number of patients to be enrolled before delivery of the appliance was increased to 70. Drop-outs are considered, due to the short follow-up time, an unlikely event.

The calculation of the sample size was performed with g * power, considering a standard deviation of 1.688 obtained from a study in the literature, with an alpha error of 0.05 and a power of 80%.

#### 2.5.1. Randomization (Random Number Generation, Allocation Concealment, and Implementation)

The randomization between the two groups’ codes was performed through a blinded random sequence of balanced blocks. This kept an equal number of patients between the two groups. The block sizes varied at random between 4, 6, and 8. A sequence of consecutively numbered and opaque envelopes contained a unique code associated with a box containing the home treatment for the patient. Assignment of the intervention: the opening of the envelope and the delivery of the container of the test or control group took place during visit III (T2) (UTO evaluation).

Patients were randomly assigned to either the test or control groups in a 1:1 ratio.

Sequence generation: the allocation sequence was generated by the Sealed Envelope application https://www.sealedenvelope.com (accessed on 1 September 2022).

#### 2.5.2. Blinding

Both patients and health care providers (i.e., data collectors) were blinded to group assignment.

### 2.6. Statistical Analysis

The quantitative variables at baseline for the test and the control group were expressed as mean with standard deviation (SD) and median with interquartile range (IQR). The quantitative outcome variables were further described with a 95% confidence interval (95% CI). The Shapiro–Wilk test was used to assess the “goodness of fit” of these variables to the normal distribution. The normality distribution requirement was not satisfied; therefore, a nonparametric test for intergroup comparison was used (U-Mann–Whitney). The qualitative variables were expressed as a frequency distribution and tested for intergroup comparison with a Chi-square test.

All statistical comparisons were conducted at the 0.05 level of significance, using R software 4.0.2 (R Foundation for Statistical Computing, Vienna, Austria).

## 3. Results

One hundred ten patients were initially screened for participation in the study. All of them accepted to participate and were enrolled. Seventy patients exhibited symptoms of UTO and attended the T0 visit. Therefore, they were randomized and considered the study sample.

The mean age of the study sample was 13 years (SD 5.6); 47.1% (*n* = 33) of the patients were women, and 52.9% (*n* = 36) were men. Baseline demographic and clinical variables are summarized in Table 1 and Table 2.

Fifty-seven patients ended the follow-up period. The dropouts from the two groups were: 8 patients (22.9%) from the test group; 5 patients (14.3%) from the control group, with no statistically significant difference between groups (*p* = 0.539). The reason for drop-outs was the patient unavailability to attend a baseline visit within 24 h starting from UTO onset. No adverse events were reported. No patients reported any problems or disorders related to the local administration, either in the test group or in the control group, such as, but not limited to, headache, effect on sleeping habits, or ulcer-bleeding release of inflammatory mediators.

In terms of painkillers taken, five patients (18.5%) in the test group and five patients (16.7%) in the control group reported they needed to take paracetamol as prescribed, without any statistically significant difference (Table 3).

The mean dimension of the lesion at T0, T1, and T2 was 6.03, 5.43, and 4.78 for the test group and 6.41, 6.27, and 5.83 for the control group, respectively (Figure 4). The inter-group comparison showed a statistically significant greater dimension of the lesion in the control group at T2 when compared to the test group (*p* value = 0.03199, test group 95%CI = 4.09–5.46; control group 95% CI = 5.08–6.59) (Table 4).

The subjective evaluation reported through the mean VAS at T0, T1, and T2, was 5.19, 2.52, and 0.44 for the test group and 5.40, 4.40, and 4.00 for the control group, respectively (Figure 5). A statistically significant difference between the test and control groups was found at both T1 (*p* value > 0.001, test group 95%CI = 2.16–2.87; control group 95% CI = 4.33–5.26) and T2 (*p* value > 0.001, test group 95%CI = 0.11–0.78; control group 95% CI = 3.35–4.65), meaning that the pain experienced by the patients belonging to the test group was significantly lower than the pain of the patients in the control group (Table 4).

## 4. Discussion

Aphthous lesion (Figure 1) is an oral mucosal inflammatory lesion that affects all different oral tasks, from feeding to speaking, damaging the quality of life (QOL) of patients, especially during orthodontic fixed therapies [11].

In the present study, the combined effect of orthodontic wax and HA plus PVP gel has been evaluated.

The primary outcome variable (VAS score at 5 days after TOU onset) demonstrated that the use of orthodontic wax in combination with BMG0722 gel (test group) can significantly reduce the perceived pain when compared with the use of orthodontic wax and a placebo (mean VAS of 0.44 and 4.00, respectively). Already three days after the TOU’s onset, there was a significant reduction in pain in the test group. The results evidenced that the use of BMG0722 gel and orthodontic wax showed a smaller dimension of the lesion after 5 days of treatment, compared to placebo gel and orthodontic wax.

Several authors recommend viscous gels to obtain a polymer film on the lesion [12], which helps pain relief during aphthous lesion formation and thus diminishes the healing time [13,14,15,16].

Similarly, a previous study evidenced the efficacy of HA in the management of Lichen Planus. The authors reported significant efficacy within the first 4 h of using HA 0.2% [17]. Another study found that twice-daily application of 0.2% HA gel had a positive effect on recurrent oral ulcers caused by Behçet disease, with a subjective reduction in the number of ulcers and an improvement in VAS for pain [18,19,20].

Bisphosphonate calcium ion depletion has been described as another factor causing mucosal and dermal ulceration with exposure of underlying bone in patients with osteoporosis. Those patients could be the target for the new formula applied to dental implants [21,22].

HA is a hygroscopic macromolecule that is highly osmotic; this property is likely to be linked with the tissue’s response to inflammation [23,24,25,26]. Savadori et al. described similar results in a double-blind randomized clinical trial with the use of 12% polyvinylpyrrolidone and 0.2% sodium hyaluronate for the treatment of minor recurrent aphthous ulcers [27].

The effects of HA gel on the oral cavity have also been documented in different clinical situations, such as the treatment of chronic periodontitis by the subgingival application after scaling and root planing, with a decrease in inflammation [28,29,30,31,32].

In conclusion, under the limitations of the study, the use of a new gel containing HA and PVP together with orthodontic wax for the treatment of TOUs promotes pain reduction and faster wound healing.

## Figures and Tables

**Figure 1 bioengineering-09-00761-f001:**
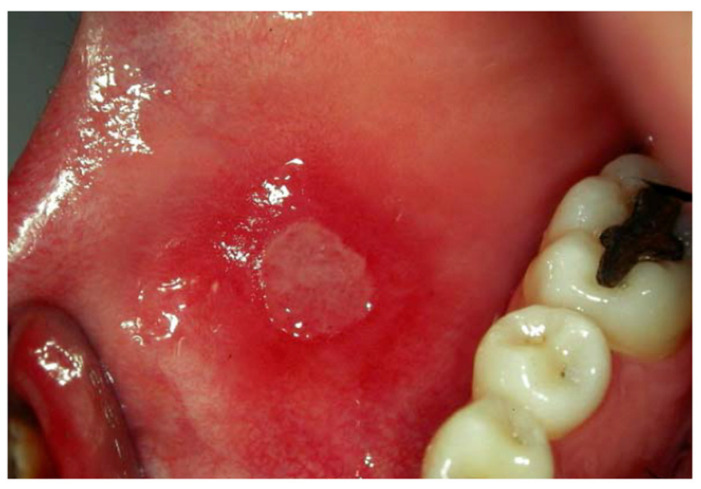
Aphthous lesion in an adult patient.

**Figure 2 bioengineering-09-00761-f002:**
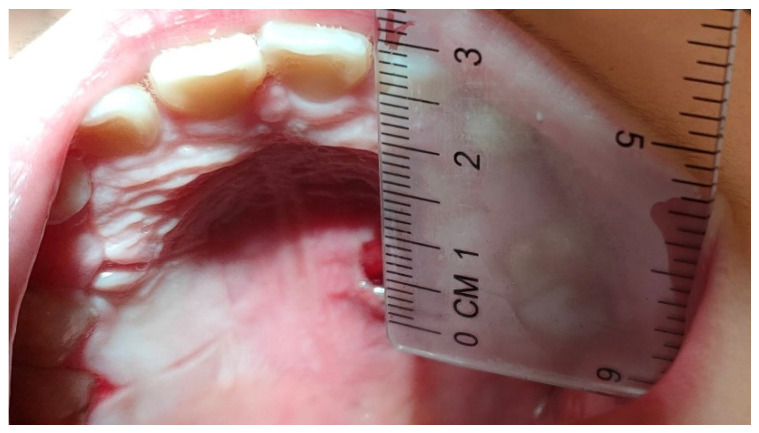
Photo Pre gel Test.

**Figure 3 bioengineering-09-00761-f003:**
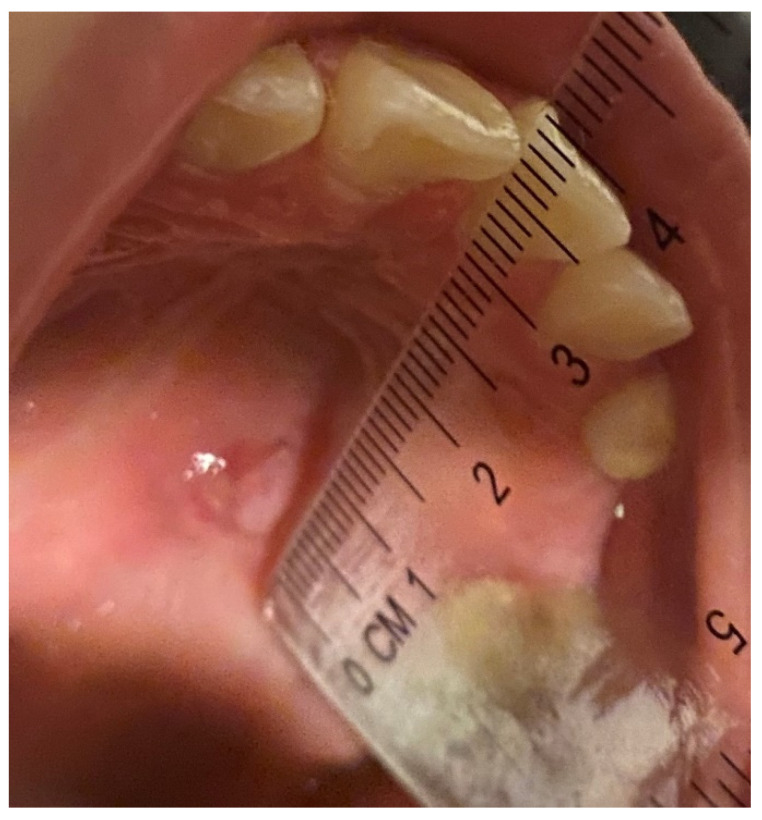
Photo Post gel Test.

**Figure 4 bioengineering-09-00761-f004:**
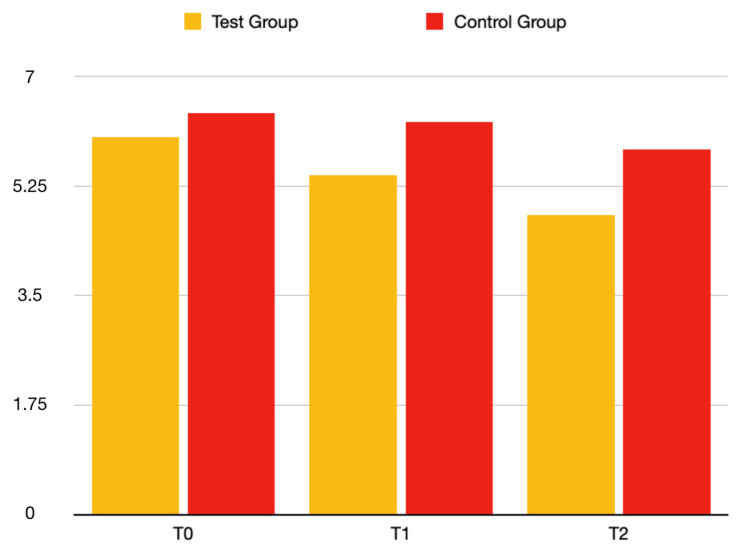
Graphic reporting the mean dimension of the lesion in the test group and control group at T0, T1, and T2.

**Figure 5 bioengineering-09-00761-f005:**
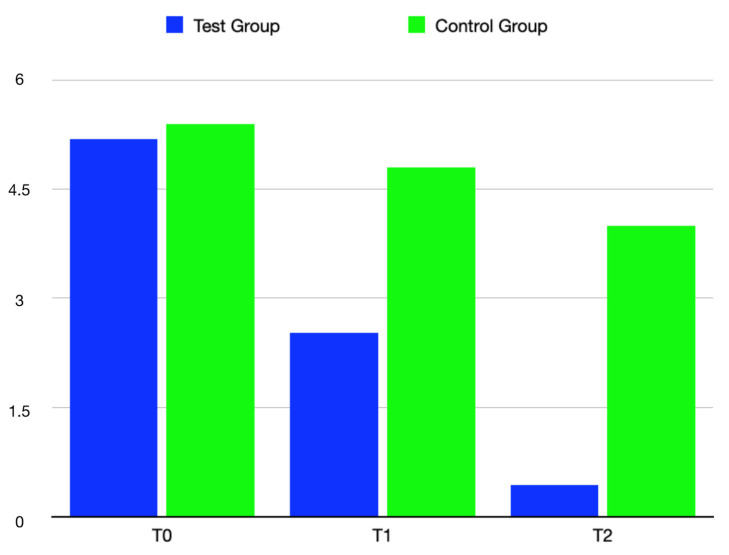
Graphic describing the mean VAS score reported by patients in the test group and control group at T0, T1 and T2.

**Table 1 bioengineering-09-00761-t001:** Demographic variables and qualitative clinical variables at baseline (T0).

		Test Group	Control Group
Total number		*n* (%)	*n* (%)
		35	35
Gender			
	Female	17 (48.6)	16 (45.7)
	Male	17 (48.6)	19 (54.3)
	Not reported	1 (2.9)	0 (0.0)
Site location			
	right hemipalate	7 (20.0)	2 (5.7)
	left hemipalate	2 (5.7)	5 (14.3)
	Right tongue	0 (0.0)	1 (2.9)
	Left tongue	2 (5.7)	0 (0.0)
	Right inner cheek mucosa	9 (25.7)	9 (25.7)
	Left inner cheek mucosa	7 (20.0)	8 (22.9)
	Right labial mucosa	4 (11.4)	5 (14.3)
	Left labial mucosa	4 (11.4)	5 (14.3)
Color			
	White	18 (51.4)	16 (45.7)
	White-red	3 (8.6)	5(14.3)
	Red	14 (40.0)	14 (40.0)
Surface			
	Endophyic	17 (48.6)	18 (51.4)
	Exo-endophytic	6 (17.1)	9 (25.7)
	Exophytic	12 (34.3)	8 (22.9)

**Table 2 bioengineering-09-00761-t002:** VAS and dimension of the lesion at baseline (T0).

	Test Group	Control Group
VAS (T0)		
Median (IQR)	4.00 (4.00, 6.00)	6.00 (4.00, 6.00)
Mean (SD)	5.19 (1.39)	5.40 (1.40)
Dimension (T0)		
Median (IQR)	6.00 (5.00, 7.00)	6.00 (5.00, 8.00)
Mean (SD)	6.03 (2.11)	6.41 (2.17)

**Table 3 bioengineering-09-00761-t003:** Drop out patients at the last follow-up visit and proportion of patient who needed to take painkillers to relief pain.

	Test Group	Control Group	*p* Value
VAS (T1)			>0.001
Median (IQR)	2.00 (2.00, 3.00)	4.00 (4.00, 6.00)	
Mean (95% CI)	2.52 (2.16–2.87)	4.80 (4.33–5.26)	
VAS (T2)			>0.001
Median (IQR)	0.00 (0.00, 0.00)	4.00 (2.00, 6.00)	
Mean (95% CI)	0.44 (0.11–0.78)	4.00 (3.35–4.65)	
Dimension (T1)			0.1223
Median (IQR)	5.00 (4.25, 6.00)	6.00 (4.50, 8.00)	
Mean (95% CI)	5.43 (4.69–6.16)	6.27 (5.49–7.05)	
Dimension (T2)			0.03199
Median (IQR)	4.50 (3.50, 5.50)	6.00 (4.50, 8.00)	
Mean (95% CI)	4.78 (4.09–5.46)	5.83 (5.08–6.59)	

**Table 4 bioengineering-09-00761-t004:** VAS and dimension of the lesion at 3 days (T1) and 5 days (T2).

	Test Group	Control Group	p Value
VAS (T1)			>0.001
Median (IQR)	2.00 (2.00, 3.00)	4.00 (4.00, 6.00)	
Mean (95% CI)	2.52 (2.16–2.87)	4.80 (4.33–5.26)	
VAS (T2)			>0.001
Median (IQR)	0.00 (0.00, 0.00)	4.00 (2.00, 6.00)	
Mean (95% CI)	0.44 (0.11–0.78)	4.00 (3.35–4.65)	
Dimension (T1)			0.1223
Median (Iqr)	5.00 (4.25, 6.00)	6.00 (4.50, 8.00)	
Mean (95% Ci)	5.43 (4.69–6.16)	6.27 (5.49–7.05)	
Dimension (T2)			0.03199
Median (Iqr)	4.50 (3.50, 5.50)	6.00 (4.50, 8.00)	
Mean (95% Ci)	4.78 (4.09–5.46)	5.83 (5.08–6.59)	

## Data Availability

Not applicable.

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
