# Peer review of "Clinical Performance Evaluation of a Hyaluronic Acid Dental Gel for the Treatment of Traumatic Ulcers in Patients with Fixed Orthodontic Appliances: A Randomized Controlled Trial"

_bioengineering, 2022, doi:10.3390/bioengineering9120761_

Round 1
Reviewer 1 Report
Orthodontic wax must still be used for the traumatizing agent to be rounded off. HA gel changes the healing time favorably. I wonder about the product's clinical value in contrast to the statistical one. Otherwise a nice conducted the study.
Reviewer 2 Report
Dear editor
Thank you for sending MS for revision
The MS discussed the effective use of hyaluronic acid dental gel for the treatment of traumatic ulcers in patients with fixed orthodontic appliances
I still have points
- I s there any other reported signs such as headache, effect on sleeping habits, reported ulcerbleeding release of inflammatory mediators.
- Is there any patient reported 1st or 2nd infection either in the control group or the treated groups.
The MS requires extensive English editing
Reviewer 3 Report
How did you sterilize the rigid plastic ruler?
Wouldn't have been more accurate to use a flexible ruler?
When referring to the concern about ulcers in the introduction, the following manuscripts can be cited:
- Evid Based Dent. 2022 Mar;23(1):40-42. doi: 10.1038/s41432-022-0250-2. Epub 2022 Mar 25. - Med Hypotheses. 2017 Sep;107:22-25. doi: 10.1016/j.mehy.2017.07.013. Epub 2017 Jul 18.I look forward to reading your corrections.
Round 2
Reviewer 2 Report
Dear authors and editors
The MS still need English editing
Author Response
English language was edited.